# Evaluation of Unsupervised Deep Learning-based Methods for Chest and Abdominal CT Image Registration

**Abstract.** Image registration is crucial for the alignment of medical images, enabling better analysis and interpretation. Although deep learning-based methods have shown promising results, the impact of architectural choices remains unclear, especially when training on scarce, small, or low-quality datasets. This study compares four different architectures for deep learning-based image registration. All methods were trained in an unsupervised setting, using the same loss function and optimization method, but with optimized hyperparameters for each method. In addition, two conventional optimization-based methods were included in the comparison. Experiments were performed on Lung and Abdomen CT datasets of the Learn2Reg challenge. Our findings suggest that the performance of deep learning-based methods varies substantially depending on the dataset type and its specific challenges.

**Keywords:** Image Registration · Deep Learning · Chest CT · Abdominal CT · Evaluation.

## 1 Introduction

Image registration is a critical process in radiotherapy and radiology. Its primary objective is to align images to achieve spatial correspondence [7,17,16]. Various studies are currently investigating image registration using deep learning (DL) with both unsupervised and supervised training approaches [19,14,11,6,15]. Jena et al. [9] evaluated the advantages and limitations of DL methods compared to classical optimization-based methods on brain MRI datasets. Their findings indicate that DL methods may outperform classical methods when sufficient labeled data are available for supervised training. However, obtaining these high-quality labels in clinical practice poses a significant challenge due to the substantial time and expertise required by clinicians [1]. Similarly, Jian et al. [10] investigated whether advanced computational modules improve the accuracy of registration on MRI datasets, suggesting that simpler designs can often achieve equal or greater performance than complex architectures. Despite extensive research on DL methods, the impact of architectural choices on performance across various organs and imaging modalities remains unclear [5], particularly when working with scarce, small, or low-quality datasets that hinder accurate registration.

To address these gaps, we present a systematic performance comparison of commonly used DL methods, including VoxelMorph (VXM) [3], Volume Tweening Network (VTN) [20], TransMorph (TSM) [4], and Contrastive Learning Registration Architecture based on VoxelMorph (CLM) [13]. As baseline comparisons, we include two widely used classical optimization-based methods, Elastix [12] and Advanced Normalization Tools (ANTs) [2]. To investigate the impact of small, low-quality datasets on model performance, our study analyze inter-patient registration on lung images and intra-patient registration on abdominal images, using two public datasets from the Learn2Reg Grand Challenge (`https://learn2reg. grand-challenge.org/Datasets/`). Given the challenge of limited labeled data, we evaluate all methods in an unsupervised setting, using overlap and deformation plausibility metrics to assess alignment accuracy and anatomical consistency.

## 2    Material and Methods

### 2.1    Image Registration Methods

Given two images, the fixed image $f(\mathbf{x})$ and the moving image $m(\mathbf{x})$, where $\mathbf{x}$ denotes the coordinate, image registration aims to enhance spatial correspondence by applying a transformation $\phi$ to the moving image. The objective is to minimize a cost function that quantifies the dissimilarity between the transformed moving image and the fixed image:

$$\hat{\phi} = \arg\min_{\phi} \left( L_{\text{sim}}\big(f, m \circ \phi\big) + \lambda \, L_{\text{smooth}}(\phi) \right) \tag{1}$$

where $m \circ \phi$ denotes the transformed moving image. The function $L_{sim}$ quantifies the similarity between the registered images, while $L_{\text{smooth}}$ regulates transformation smoothness, with $\lambda$ as the trade-off parameter for regularization strength.

**Classical Image Registration.** The classical registration process between $f$ and $m$ iteratively optimizes transformation parameters to minimize a similarity metric as given in Eq 1. Elastix and ANTs were employed as classical optimization-based methods. For ANTs, we used Symmetric Normalization (SyN) for affine plus deformable registration with mutual information as the optimization metric. In Elastix, affine and B-spline transformations were used for affine and deformable registration, optimizing mutual information via adaptive stochastic gradient descent.

**Deep Learning-Based Image Registration.** DL approaches are designed to directly predict displacement fields, typically using an encoder-decoder architecture. Variations among these methods lie primarily in how they generate the displacement field and the resulting transformed image. We examined VXM, VTN, TSM, and CLM to investigate how diverse architectures could improve performance (Fig. 1). Although these methods support supervised training, we focused on their unsupervised learning form. VXM employs a convolutional neural network to predict a dense displacement field that aligns $m$ to $f$ using a spatial

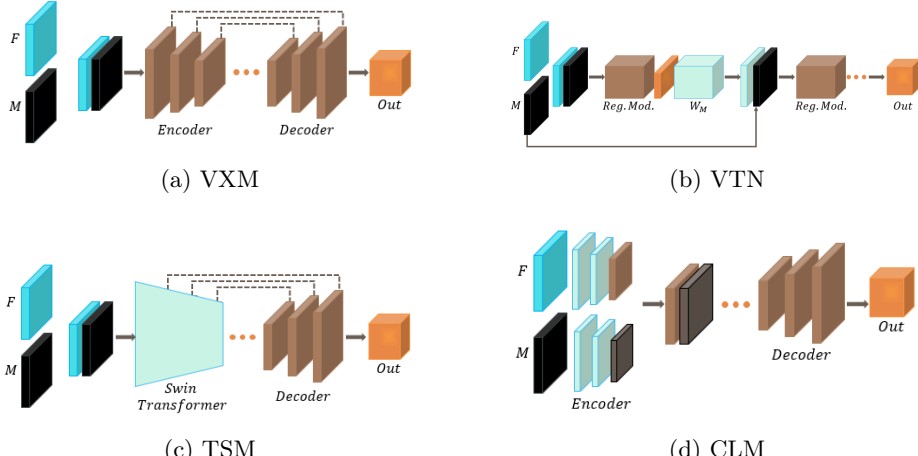

(a) VXM                    (b) VTN

(c) TSM                    (d) CLM

Fig. 1: Deep learning-based image registration methods.

transformer network[8]. VTN employs a recursive registration strategy, repeating the registration process (VXM) three times, while TSM leverages transformers to enhance feature extraction. Unlike the other methods, CLM employs a dual-encoder structure to process $f$ and $m$ independently and subsequently uses a decoder architecture analogous to the other methods. All these methods first apply an initial affine transformation, followed by deformable registration. For the complete transformation, the total loss $L_{\text{total}}$ as the sum of the affine transformation loss $L_{\text{aff}}$ and the deformable transformation loss $L_{\text{def}}$ is used:

$$L_{\text{total}} = \underbrace{\sum_{i=1}^{3}(\sigma_i^2 + \sigma_i^{-2}) + \det(\mathbf{A} + \mathbf{I})}_{L_{\text{aff}}} + \underbrace{\text{Corr}(f, m \circ \phi) + \lambda \sum_{\mathbf{x} \in \Omega} \|\nabla u(\mathbf{x})\|^2}_{L_{\text{def}}} \quad (2)$$

The $L_{\text{aff}}$ is the combination of a orthogonality loss and determinant loss to penalize extreme affine transformations [20], where $\sigma_i$ are the singular values of the affine matrix $\mathbf{A}$, and $\mathbf{I}$ denotes the identity matrix. The first term of $L_{\text{def}}$ represents the Pearson correlation coefficient as the similarity metric, assessing the alignment of the images, while the second term employs total variation loss for regularization, promoting smoothness in the displacement field $u(\mathbf{x})$.

## 3 Experiments and Results

### 3.1 Datasets

**Lung CT Dataset -** It includes CT scans from 30 patients, divided into 20 training and 10 testing cases. Each case contains scans from both the inspiration and expiration phases. Each scan has dimensions of $192 \times 192 \times 208$, with a voxel

spacing of $1.75 \times 1.25 \times 1.75$ mm. The training dataset includes lung segmentations and keypoints, while the testing dataset provides lung segmentations and annotated landmarks. The expiration scan is set as the fixed image, and the inspiration scan as the moving image.

**Abdomen CT Dataset -** It includes 30 training and 20 testing scans, with fixed and moving images randomly selected during training from different patients. Each scan has dimensions of $192 \times 160 \times 256$, with an isotropic voxel spacing of 2 mm. The training dataset includes 13 manual anatomical segmentations per patient. For the testing dataset, no manual segmentations were provided; therefore, liver segmentations were generated using TotalSegmentator [18] and subsequently validated and corrected by a radiologist when necessary.

Both datasets pose significant challenges, including large deformations, small sample sizes, and partial organ visibility. Scan fields of view are inconsistent. Lung scans may miss parts of the lungs due to respiration, while abdominal scans can include varying portions of lower thoracic organs. Manual annotations, including segmentations, landmarks, and keypoints, were used exclusively for evaluation purposes, as the methods operate in an unsupervised learning setting.

### 3.2    Implementation Details

We initialized the network weights using the Xavier initialization method with a uniform distribution and optimized them using the Adam optimizer ($\beta_1 = 0.5$, $\beta_2 = 0.999$), learning rate set to $1e^{-4}$ with a weight decay of $1e^{-4}$. The learning rate scheduler had a step size of 10 and a $\gamma = 0.96$. A batch size of 1 was established during training, conducted for 100 epochs, since all methods had converged by that point. For ANTs, we utilized default parameters, including iterations set to $\{100,100,70\}$, a convergence threshold of $1 \times 10^{-6}$, smoothing factors of $\{0, 1, 2\}$, shrink factors of $\{4, 2, 1\}$, and a regularization weight ranging from 0.01 to 0.1. Similarly, we applied the default parameters for Elastix, using a grid spacing of 8 voxels (lung) and 32 voxels (liver) to balance flexibility and smoothness. Multi-resolution registration was implemented across three levels. The code for this study is available on GitHub at `https://github.com/research-medical-imaging/image-registration`.

### 3.3    Evaluation Metrics

We evaluated performance using the Dice Similarity Coefficient (DSC), 95th percentile Hausdorff Distance (HD), Average Surface Distance (ASD), proportion of voxels with non-positive Jacobian determinant ($J < 0$), and the standard deviation of the Jacobian determinant ($\text{std}(J)$), capturing both overlap and deformation plausibility and physical realism through Jacobian-based metrics. For the lung dataset, Target Registration Error (TRE) was included based on annotated landmarks. HD, ASD, and TRE are reported in voxels. Statistical significance was assessed using pairwise Wilcoxon signed-rank tests.

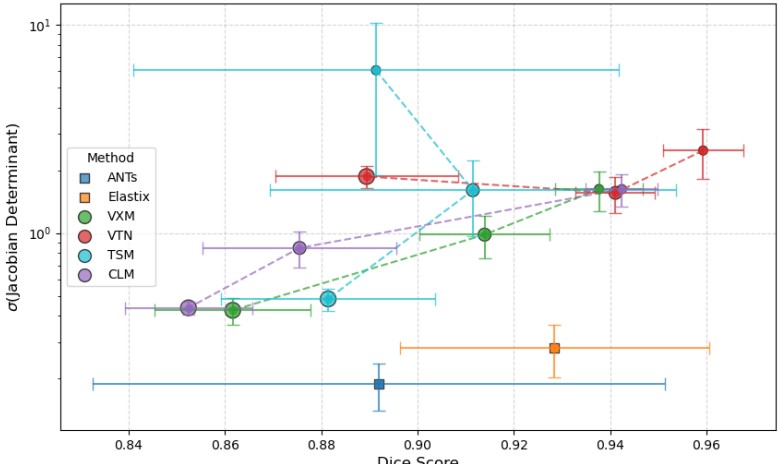

(a) Results on the Lung CT dataset.

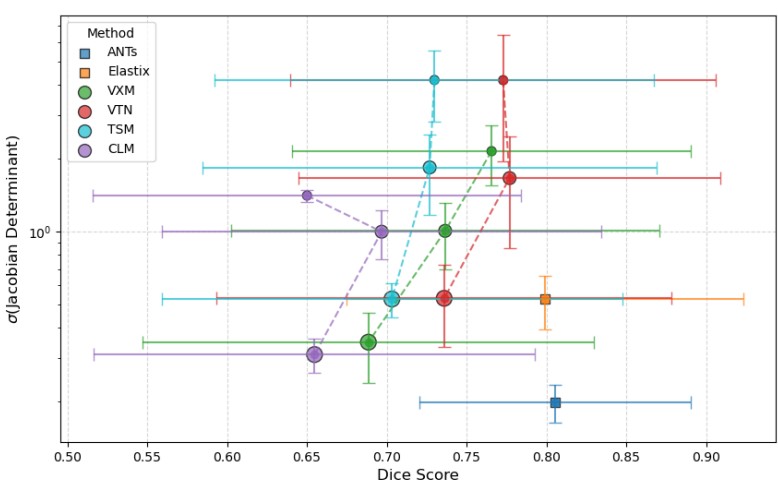

(b) Results on the Abdomen CT dataset.

Fig. 2: Scatter plot of DSC versus std($J$) for each method across three values of $\lambda$, which controls the smoothness of the deformation field as seen in Eq. 2. Circle size indicates the value of $\lambda$: small (0.1), medium (1.0), and large (10.0).

### 3.4   Results

**Hyperparameter Optimization -** To optimize the hyperparameter $\lambda$ in the total loss function with respect to DSC, we evaluated values of 0.1, 1.0, and 10.0. On the Lung dataset, most methods achieved their highest DSC at $\lambda = 0.1$, with the exception of TSM ($\lambda = 1.0$). On the Abdomen dataset, $\lambda = 1.0$ yielded optimal performance for all methods except VXM ($\lambda = 0.1$). To further investigate the trade-off between segmentation accuracy and deformation regularity, Fig. 2 illustrates scatter plots of DSC versus std($J$), across different $\lambda$ values. The y-axis uses a logarithmic scale to accommodate the broad range of deformation variability. Notably, TSM and VTN exhibit increased std($J$) at $\lambda = 0.1$ on both Lung and Abdomen datasets, respectively, indicating more irregular and potentially less stable deformation fields under these settings.

**Time of Inference and Memory Usage -** Once trained, DL methods offer significantly faster inference times compared to classical optimization-based approaches. Among all evaluated methods, CLM achieved the fastest runtime, while Elastix was the slowest. In terms of memory usage, VTN required the most memory, whereas CLM was the most memory-efficient. Detailed runtimes and memory usage are as follows: ANTs (105s), Elastix (129s), VXM (0.20s, 309MB), VTN (0.14s, 620MB), TSM (0.31s, 451MB), and CLM (0.10s, 230MB).

**Quantitative Results -** Tables 1 and 2 present the quantitative results for the Lung and Abdomen CT datasets, respectively. Statistical significance was assessed using ANTs as the reference method, with a standard threshold of $p < 0.05$. Significant differences are indicated with an asterisk (*).

Table 1: Results for the Lung CT dataset.

| Method | DSC ↑ | HD ↓ | ASD ↓ | TRE ↓ | $J < 0$ ↓ | std($J$) ↓ |
|--------|-------|------|-------|-------|-----------|------------|
| ANTs | 0.89±0.06 | 15.30±9.39 | 2.91±1.95 | **6.75±2.96** | **0.00±0.00** | **0.19±0.05** |
| Elastix | 0.93±0.03 | 10.53±8.20* | 2.07±1.60 | 6.95±2.85* | **0.00±0.00*** | 0.28±0.08* |
| VXM | 0.94±0.01 | 4.40±0.64* | 1.07±0.19* | 7.16±2.97 | **0.00±0.00*** | 1.63±0.36* |
| VTN | **0.96±0.01*** | **2.53±0.76*** | **0.66±0.17*** | 7.11±2.82 | 0.08±0.03* | 2.49±0.67* |
| TSM | 0.91±0.04 | 11.24±7.63 | 2.09±1.55 | 7.20±2.97* | 0.07±0.01* | 1.61±0.64* |
| CLM | 0.94±0.01 | 4.53±0.56* | 0.96±0.14* | 7.26±2.95* | 0.11±0.03* | 1.62±0.28* |

On the Lung dataset (Table 1), VTN achieved the highest DSC ($0.96 \pm 0.01$), along with the lowest HD ($2.53 \pm 0.76$) and ASD ($0.66 \pm 0.17$), all significantly better than ANTs. Although ANTs yielded the lowest mean TRE ($6.75 \pm 2.96$), the difference with VTN was not statistically significant ($p > 0.05$), indicating comparable target localization accuracy. VXM also demonstrated competitive performance, where it outperformed ANTs in both HD and ASD. In contrast,

Table 2: Results for the Abdomen CT dataset.

| Method | DSC ↑ | HD ↓ | ASD ↓ | $J < 0$ ↓ | std($J$) ↓ |
|--------|-------|------|-------|-----------|------------|
| ANTs | **0.81±0.08** | 17.28±8.85 | **3.51±1.97** | **0.00±0.00** | **0.20±0.03** |
| Elastix | 0.80±0.12 | 17.52±9.85 | 4.43±4.04[*] | 0.02±0.02[*] | 0.53±0.13[*] |
| VXM | 0.77±0.12[*] | **16.80±9.16** | 4.37±3.59[*] | 0.02±0.01[*] | 2.14±0.59[*] |
| VTN | 0.78±0.13[*] | 17.06±10.01 | 4.11±4.10 | 0.02±0.01[*] | 1.66±0.80[*] |
| TSM | 0.73±0.14[*] | 18.32±10.11 | 5.05±4.52[*] | 0.05±0.01[*] | 1.83±0.66[*] |
| CLM | 0.70±0.14[*] | 18.91±10.22[*] | 5.37±4.47[*] | 0.05±0.01[*] | 1.00±0.23[*] |

results on the Abdomen dataset (Table 2) consistently favored ANTs, which achieved the highest DSC ($0.81 \pm 0.08$), lowest ASD ($3.51 \pm 1.97$), and regular deformations. DL-based methods exhibited statistically significant degradation in most metrics. TSM and CLM showed particularly high values in terms of $J < 0$ and std($J$), indicating irregular deformation fields. Although VXM achieved the lowest HD ($16.80 \pm 9.16$), this improvement was not significant ($p > 0.05$). VTN performed relatively well in DSC and HD, but still showed worse regularity than ANTs.

**Qualitative Results -** Fig. 3 presents a qualitative comparison of results across both datasets. For the Lung CT dataset, deformed moving images and segmentations appear visually consistent across all methods, but differences emerge in the Jacobian determinant maps. Classical methods exhibit smooth and regular deformation fields, while DL methods, particularly VTN and CLM, show irregular patterns, indicating non-smooth or non-invertible transformations. For the Abdomen CT dataset, all methods produce anatomically implausible deformations, with DL methods demonstrating greater irregularities in the Jacobian determinant maps.

## 4   Discussion and Conclusion

In this study, we focused on registration performance under the constraints of small datasets and significant anatomical variability, conditions that reflect many real-world clinical challenges. Our observations highlight several key limitations and trade-offs in current DL-based registration methods in such settings.

A key observation is the sensitivity of DL methods to hyperparameter selection. Performance variability in terms of DSC underscores the need for careful hyperparameter optimization. Based on the Wilcoxon test, DL-based methods significantly outperformed ANTs on the Lung dataset across DSC, HD, and ASD, with VTN leading the results. However, on the Abdomen dataset, ANTs remained the most stable and accurate method across all metrics. Despite the strong quantitative performance of VTN, it produced anatomically implausible deformations, as evidenced by irregular Jacobian determinant patterns, partic-

| Fixed | Moving | ANTs | Elastix | VXM | VTN | TSM | CLM |
|-------|--------|------|---------|-----|-----|-----|-----|

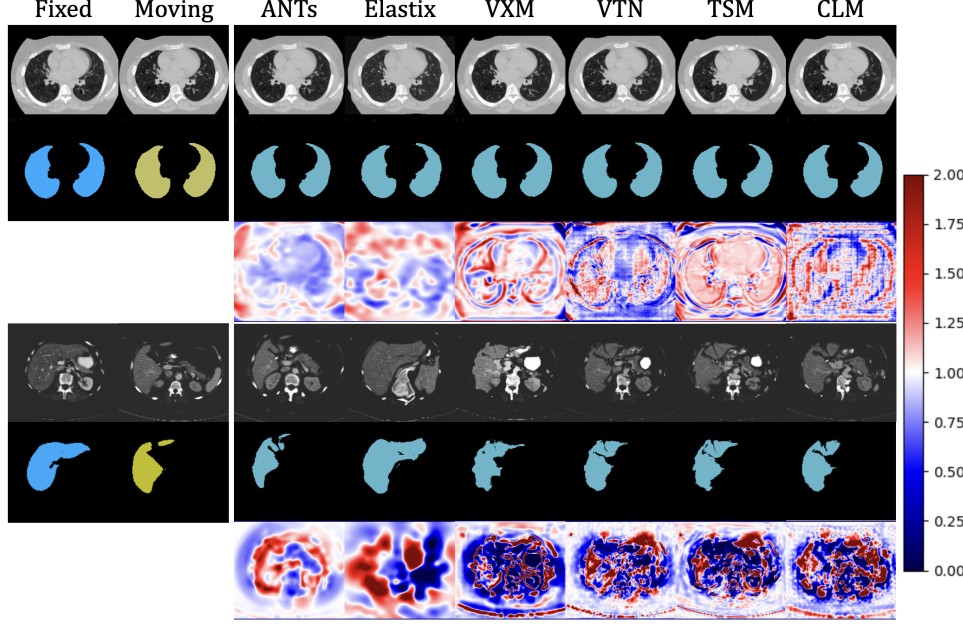

Fig. 3: Qualitative comparison. First two columns show fixed and moving images with segmentations. Rows 1–2: deformed image and segmentation; row 3: Jacobian map. Top: Lung CT; bottom: Abdomen CT.

ularly in the Abdomen CT dataset, which may be attributed to the increased challenge of inter-patient variability. DL methods introduced non-physical distortions in soft tissue regions, whereas classical approaches like ANTs generated smoother, anatomically more realistic Jacobian determinant maps. These findings highlight the risks of relying solely on overlap-based metrics such as DSC to evaluate registration quality. Analysis of the correlation between DSC and $\text{std}(J)$ (Fig. 2) revealed that higher overlap scores did not consistently align with physically plausible deformations, raising concerns about clinical applicability. Additionally, convolution-based models (VXM) consistently outperformed transformer-based models (TSM) across both datasets, with VXM offering better memory efficiency and performance. This supports recent findings that simpler architectures can match or exceed more complex models [10]. In general, our results highlight persistent challenges in DL-based registration with limited data and high anatomical variability. They emphasize the need for task-specific strategies, anatomically plausible regularization, and evaluation frameworks that go beyond overlap metrics. Future work should standardize hyperparameter optimization, incorporate anatomical priors, and ensure validation across diverse anatomies, modalities, and pathologies. Robust registration must ensure not only accurate overlap but also anatomically plausible and clinically interpretable transformations, even under data or resource constraints.

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
