# OpenReview forum: "Evaluation of Unsupervised Deep Learning-based Methods for Chest and Abdominal CT Image Registration"
_MICCAI.org/2025/Workshop/MSB_EMERGE — Submitted to MSB EMERGE 2025_

### Official Review · Reviewer_f1Cc · 2025-07-09

**Recommendation:** 2
**Confidence:** 4

**Clarity:**

The paper is generally clear but has some clarity issues that could be addressed with moderate revision

**Feedback:**

* To support the generality of the findings, it would be helpful to include evaluations on additional datasets.
* Figure 2 is somewhat difficult to interpret at first glance; a tabular representation might convey the comparison more clearly.
* If the goal is to study registration under limited data, proposing a method that addresses data scarcity (e.g., a data augmentation strategy or a lightweight framework) could strengthen the contribution.

**Justification:**

The paper provides a useful comparison of existing methods, but lacks novelty and broader validation. With further development (especially in addressing the low-data challenge more directly), the work could offer more substantial contributions.

**Reproducibility:**

Sufficient amount of details available for reproducing the main results, and open access is provided (or promised upon acceptance) to source code and/or data

**Strengths:**

* The authors establish a well-controlled experimental setting, enabling fair comparisons across multiple registration methods.
* The background for registration are clearly explained, making the paper accessible and easy to follow.

**Summary:**

This paper conducts a comparative study of deep learning-based registration architectures under limited data conditions.

**Weaknesses:**

* The work mainly focuses on benchmarking existing architectures, and does not introduce new methodological contributions.
* The motivation for studying deep-learning-based registration in low-data regimes could be more clearly justified. Without broader validation, the reported results may reflect overfitting to a specific dataset.

---

### Official Review · Reviewer_Adxn · 2025-07-09

**Recommendation:** 2
**Confidence:** 4

**Clarity:**

The paper is clear and well-written, with minor areas for improvement in clarity

**Feedback:**

Aside from addressing the points above in Weaknesses, especially about increasing the scale of the analysis, please also note the following:
1. Please improve the formatting of Table 1. The VTN row, for example, is quite cramped and hard to read.
2. For the statistical tests, consider reporting significance at multiple levels using the standard notation of *, **, ***. This would allow more insight than the current single significance level testing.
3. Please proofread the paper. Although it is mostly well written, I found at least one typo. (page 2) "our study analyze" should be "our study analyzes".

**Justification:**

This paper would be useful for people interested in medical image registration, but because of the relatively small scale of analysis, it is unclear how general the takeaways are.

**Reproducibility:**

Sufficient amount of details available for reproducing the main results, and open access is provided (or promised upon acceptance) to source code and/or data

**Strengths:**

1. The authors evaluate multiple classical optimization-based and DL-based medical image registration algorithms on 2 CT datasets.
2. Extensive experiments, hyperparameter optimization, and statistical testing help support the sound-ness of the observations.
3. The paper is generally well written, including the methodological formulations, and is easy to follow for most parts.

**Summary:**

The authors conduct an analysis of 2 optimization-based and 4 DL-based unsupervised medical image (CT) registration.

**Weaknesses:**

1. The paper is intended to be an analysis paper for comparing/benchmarking existing methods. Such papers are quite useful, but the scale of analyses, both in #datasets (2), #modalities (1), and #methods (6) is limiting, meaning it is not clear how generalizable the observations from this analysis would be.
2. I find this statement to be incorrect/contentious: "Once trained, DL methods offer significantly faster inference times compared to classical optimization-based approaches." for 2 reasons:
(a) Optimization-based approaches do not have a training phase, and therefore, for a fair comparison, their inference time should ideally be compared to the training+inference time of DL-based approaches.
(b) The authors do not report the memory usage for optimization-based methods (page 6). This is misleading because DL methods typically have higher memory and hardware requirements (GPU) for the reported inference times. Not specifying the hardware accelerator (GPU) leads to an unfair comparison.
3. I am not sure how to interpret Fig. 2 as it's quite cluttered. If you want to show a scatter plot, the error bars can be skipped and you can simply plot the points.

---

### Official Review · Reviewer_n4Ar · 2025-07-09

**Recommendation:** 2
**Confidence:** 4

**Clarity:**

The paper is generally clear but has some clarity issues that could be addressed with moderate revision

**Feedback:**

General feedback:

- To explore the effect of architectural choices on registration, more is needed than just a comparison of overlap and deformation metrics for four methods. A more insightful approach would be, e.g., an ablation study, which could clearly show which parts of the architecture affect the registration
- A potentially more interesting approach than just comparing 4 methods would be to, e.g., focus on the problems with DICE vs plausibility and clinical applicability as mentioned in the discussion


Things that need clarification or improvement:
- “The training dataset includes lung segmentations and keypoints, while the testing dataset provides lung segmentations and annotated landmarks.”
    - The training and testing datasets include the same type of keypoints - it is not clear what difference the authors imply between “keypoints” and “annotated landmarks.”
- “VTN employs a recursive registration strategy, repeating the registration process (VXM) three times.”
    - What is meant by the VXM in brackets?
- Fig 2. is hard to read - too much information is presented in one graph
- Fig 2. “On the Abdomen dataset, λ = 1.0 yielded optimal performance for all methods except VXM (λ = 0.1)”
    - But from the figure, it seems that TSM also achieved the best for 0.1
- Fig 2. “Notably, TSM and VTN exhibit increased std(J) at λ = 0.1 on both Lung and Abdomen datasets, respectively, indicating more irregular and potentially less stable deformation fields under these settings.”
    - It looks like all methods exhibit this increase, which makes sense because the less you regularise the deformation field, the less smooth it gets
- “Tables 1 and 2 present the quantitative results for the Lung and Abdomen CT datasets, respectively.”
    - For which hyperparameters?
- “For the Abdomen CT dataset, all methods produce anatomically implausible deformations.”
    - What is the justification? This could be the same case for the lung Jacobian maps. Only because the lung maps are consistent across methods doesn’t imply that they are plausible
- It would make sense to present the Jacobian determinant map with an included negative range, as negative values show folding; now we just see expansion and shrinking (det<0 folding, 0<det<1 shrinking, det>1 expansion)
- “convolution-based models (VXM) consistently outperformed transformer-based models (TSM) across both datasets”
    - But in the TSM paper, they consistently outperformed Voxelmorph. Why is that?

Things that are incorrect

- “Although these methods support supervised training”
    - VXM doesn’t support supervised training. Supervised training would imply a known deformation field used as ground truth (which doesn’t exist in registration, unless you, e.g., take one from a conventional method or generate it synthetically), whereas VXM allows for weakly supervised training with, e.g., segmentations.
        - quote from the VXM paper: “In the first (unsupervised) setting, we train the model to maximize standard image matching objective functions that are based on the image intensities. In the second setting, we leverage auxiliary segmentations available in the training data.”
    - VTN doesn’t support supervised (or even weakly-supervised training)
        - quote from the VTN paper: “In this article, we propose a new unsupervised learning method”
    - CXM doesn’t support supervised
        - quote from the CXM paper: “We display the overall workflow of our unsupervised segmentation technique.”
    - TSM doesn’t support that
        - quote from the TSM paper: “In this paper, we introduced TransMorph, a novel model for unsupervised deformable image registration”
- “All these methods first apply an initial affine transformation, followed by deformable registration.”
    - VoxelMorph doesn’t do that - it uses affinely preregistered images for training. You can also see in the architecture of VXM that there is no affine transformation applied to the images
        - quote from the VXM paper: “We carry out standard preprocessing steps, including affine spatial normalization.”
    - VTN does that
        - quote from the VTN paper: “We propose three innovative technical components: (1) An end-to-end cascading scheme that resolves large displacement; (2) An efficient integration of affine registration network”
    - CXM probably doesn’t do that as it is based on VXM, also, affine registration isn’t mentioned in the paper
    - TSM does that
        - quote from the TSM paper: “A novel Transformer-based neural network, TransMorph, was proposed for affine and deformable image registration”

**Justification:**

Even if all the comments were to be adressed, the paper still wouldn’t have presented any new idea, evaluation technique or novel insights. While the paper, with the mentioned improvements implemented, would be a small and clean comparison of a few methods, it still wouldn’t have shown more than evaluations from other papers (including from the papers that were evaluated here).

**Reproducibility:**

Sufficient amount of details available for reproducing the main results, and open access is provided (or promised upon acceptance) to source code and/or data

**Strengths:**

1. The paper provides a clear and simple comparison of 4 DL architectures for registration, which is supported by a good figure (Fig. 1). This figure enables the reader to quickly compare the architectures, which is usually cumbersome as every paper uses different schemes to present the architectures.

2. The paper presents a good statistical analysis, which is often lacking in other papers. This is especially important as, e.g., the DICE scores often differ by only a few hundredths, so we get a simple tool to see which differences are significant.

3. The paper highlights that the use of metrics such as DICE might be misleading and that there is no clear path from such metrics to clinical usefulness. While this is not a novel discovery, it is good to see it emphasised.

**Summary:**

The authros present a concise but limited comparison of few registration methods with a clear statistical evaluation.

**Weaknesses:**

1. A few statements about the presented figures and results seem unjustified, such as stating which deformation fields are plausible and which are implausible. Figures 2 and 3 could also be changed to improve readability. Furthermore, some statements or findings need further explanation, such as their finding that VXM outperforms TSM, while in the TSM paper, TSM outperforms VXM.
2. The paper also contains two errors in the descriptions of the methods that are compared. Firstly, stating that all four methods support supervised training, when in fact none of them support that (one or more may support weakly supervised training). Secondly, stating that all four methods first apply an affine registration, when only VTN and TSM do that (and in TSM, it is optional)
3. The authors claim to “present a systematic performance comparison of commonly used DL methods” and that they want to study the impact of architectural choices on image registration. However, they only compare 4 DL methods, perform a minimal hyperparameter search, and evaluate on two small datasets. After reading this paper, it is unclear how architectural choices impact the registration performance. Furthermore, the performance comparison is quite small, as e.g., the TransMorph method, which was evaluated in this paper, uses 4 conventional and 4 DL methods to compare in its original paper.